# Mind training, stress and behaviour—A randomised experiment

**Yonas Alem[1], Hannah Behrendt[2], Michèle Belot[3]\*, Anikó Bíró[4]**

**1** Department of Economics, University of Gothenburg, Gothenburg, Sweden, **2** Behavioural Insights Team, London, United Kingdom, **3** Cornell University, Ithaca, NY, United States of America, **4** Centre for Economic and Regional Studies, Békéscsaba, Hungary

\* mb2693@cornell.edu

## Abstract

In this paper, we evaluate the effects of a psychological training, called Mindfulness-Based Stress Reduction (MBSR) on stress and risk and time preferences. MBSR is a well-known psychological technique, which is believed to improve self-control and reduce stress. We conduct the experiment with 139 participants, half of whom receive the MBSR training, while the other half are asked to watch a documentary series, both over 4 consecutive weeks. Using a range of self-reported and physiological measures (such as cortisol measures), we find evidence that mindfulness training reduces perceived stress, but we only find weak evidence of effects on risk and inter-temporal attitudes.

## Introduction

Economists have devoted a fair amount of attention to understanding choices that are inter-temporal and uncertain in nature, such as choices on how much to save, whether to apply to college, whether to smoke or not, go on a diet, etc. There is now ample evidence that a broad range of personality traits and/or behavioral attitudes (so-called non cognitive skills) matter a lot in these decisions (see for example ([1–3] for education decisions and [4] for health).

Recent work in behavioural economics has shown that inter-temporal choices exhibit a range of "biases" such as biases toward immediate gratification. These biases can explain inconsistencies in people's choices over time, for example choosing to go on a diet and then not sticking to it ([5, 6]). One important question is: What policy interventions should we design now that we know that these traits matter. For example, if patience and self control are associated with better outcomes in life, what interventions could we think of to help those who lack patience and self control?

Policies inspired by insights from behavioural economics take behavioural traits and biases as given and often propose tools that seek to exploit these behavioural biases to the advantage of individuals. For example, by allowing workers to pre-commit to later increases in their savings rate for retirement, [7] exploit workers' present bias to help those who would like to save more but lack the willpower to do so. There has, however, been relatively little interest in whether one could actually try to correct for these biases or even try to alter the decision-making processes that underpin these behaviours in the first place. There is however a wider

**Data Availability Statement:** The data is available in the Ecommons data repository of Cornell University: https://doi.org/10.7298/s0d2-h158.

**Funding:** The study received financial support from the Swedish Research Council for Sustainable

Development (Formas), through the project "Cooperation for sustainable resource utilization", grant number: 253-210-32. Aniko Bìró acknowledges funding from the Momentum program of the Hungarian Academy of Sciences. The funders played no role in the study besides funding.

**Competing interests:** The authors have declared that no competing interests exist.

literature in psychology, documenting the effects of various activities on executive functions and self-control in particular. Reviewing a range of interventions, [8] find evidence of various such activities improving children's executive function, including aerobics, martial arts, yoga, and mindfulness. But there has been only limited research into whether these techniques could be used to affect fundamental preferences such as risk and time preferences.

In this paper, we investigate the possibility of altering behavioural preferences by training. A recent series of studies in economics ([9, 10]) shows that traits such as patience and grit are malleable in childhood and can be trained, at least early on in life. There are also a number of studies that examine the effects of "shocks" in the environment, such as natural catastrophes, on risk preferences, finding supportive evidence that these preferences may change in response to external shocks ([11, 12]). Thus, there appears to be scope for preferences to change.

This study focuses on adults. We selected a well-known psychological technique, which is based on what is referred to as "Mindfulness" training. The technique is based on repeated exercises of meditation, breathing and yoga, and has recently enjoyed a rise in popularity in many countries. There is a widely held belief that mindfulness may affect behavioural attitudes and traits, such as for example patience and self control. To our knowledge there is no evidence of the effects of these techniques on preferences or behavioural anomalies. So far, experimental studies have primarily evaluated its effects on stress and anxiety reduction (see [13–15]).

A growing body of research finds that mindfulness, especially mindfulness-based stress reduction (MBSR) and mindfulness-based cognitive therapy (MBCT) are effective treatment for health problems such as recurrent depression ([16, 17]) and anxiety ([18]). [19] study the impact of the online mindfulness course we use in the present study and find significant reductions in perceived stress, anxiety and depression at course completion, as well as a further decline at a one-month follow-up. The authors report effects that are comparable to those found in studies using face-to-face mindfulness courses and other types of treatment for stress, such as cognitive behavioural therapy.

A number of studies have found effects on executive function, including self-control ([20–23]). Because of its potential effect on stress, mindfulness practice could also affect risk and time preferences via its effect on stress. Stress triggers a physiological response in the body and neuroscientists typically distinguish between short-run (acute) and long-run (chronic) stress. The effects of acute and chronic exposure to cortisol, the primary stress hormone, can be very different, and are in many cases opposite. Most studies focus on the effects of *acute* stress on decision-making.

The evidence on causal effects of acute stress on risk-taking is mixed. A number of studies find that fearful emotions increase risk aversion ([24, 25]), while other studies find that elevated cortisol is associated with more risk-taking ([26, 27]). Other studies, such as [28], have found that, in stressful situations, humans are likely to fall back on automatized reactions to risk. Regarding time preferences, a recent study by [29] finds that temporarily elevated cortisol induces people to prefer more immediate rewards to delayed rewards.

There are a few studies that look at the effects of chronic stress, indicated by the presence of elevated cortisol levels over longer periods of time. Chronic stress has been found to impair behavioural flexibility and attentional control [[30–32]]. A recent experiment (double-blind and placebo controlled) raised cortisol levels of volunteers over a period of eight days to mimic the biological effects of chronic stress, replicating levels previously observed in real financial traders [33]. The study found that raised cortisol levels led volunteers to become more risk-averse and that men, relative to women, increasingly over-weighed small probabilities.

We conducted a randomised controlled experiment where we treated a sub-group of participants with an online course in mindfulness called "Be Mindful". We refer to the website www.

bemindfulonline.com for a detailed description of mindfulness-based stress reduction course. The course is designed as a complete training for mindfulness and is currently one of the most popular online tools for learning mindfulness skills. It is run by the UK Mental Health Foundation.

Our sample consists of 139 students from the University of Edinburgh who participated in a six-week study (with a five-month follow-up). Students were allocated randomly either to a mindfulness-based stress reduction (MBSR) programme or to a control intervention consisting of a series of documentaries called "BBC Ancient Worlds". We chose this intervention for the control group because it requires a similar degree of time commitment, but involves very different activities. While mindfulness consists of exercises that should help individuals take charge of their thought processes (becoming "mindful"), a TV documentary is more likely to be distracting. Both protocols were to be followed outside the laboratory and lasted for four consecutive weeks, starting in the week immediately after the initial session. Participants were asked to return to the laboratory for five consecutive weeks after the initial session (including one week after the interventions ended) and provide feedback on the previous week (in particular about their engagement with the intervention). We also conduct an additional post-intervention session five months later to document their longer-term behaviour and see whether there was evidence of long-term behavioural changes.

Using student subjects to answer these research questions provides more than logistical advantages over other subjects. There is strong evidence that students suffer from chronic stress [34, 35] and are particularly prone to engage in unhealthy risky behaviours such as smoking, drinking and eating unhealthy food [36–38].

The first outcome of interest is to what extent participants engaged with each of the programmes. Because both programmes require some form of commitment, one might expect, for example, that impulsiveness and present bias could correlate with the ability to complete the programme. We find that indicators of stress and behavioural attitudes are not systematically related to the likelihood of completing the mindfulness course.

We then proceed to evaluate the effects of the MBSR programme on two sets of outcome variables. We evaluate effects on measures of chronic stress, because this outcome has been a primary target of the MBSR programme, and could be one of the channels through which risk and time preferences are altered. We then study impacts on risk and inter-temporal attitudes, which are believed to play a key role in a range of decision domains such as saving, education and health.

Results suggest that compared to the control group, participants in the treatment group exhibit a consistently lower level of stress as measured by the Perceived Stress Scale (PSS). The effect remained strong five months after the end of the intervention. Turning to attitudes towards risk and time, we find suggestive evidence that participants in the treatment group are more patient, more risk-averse, less likely to suffer from present bias, although some of these effects are not statistically significant. Overall, our results suggest that mindfulness is effective at reducing stress, but the evidence on whether it can alter fundamental attitudes towards risk and time is only suggestive.

Of course, the intervention we looked at was relatively short in duration (four weeks). Nevertheless, we believe this is an important research agenda that deserves attention by economists and behavioural economists, and more research is needed to understand the extent to which it is possible to train the mind to overcome behavioural biases. Not only is the question of whether preferences and behavioural biases are malleable, but also whether there is scope for training at different stages of life.

The rest of the paper is structured as follows. Section 2 outlines Materials and Methods. Section 3 presents the Results and Section 4 concludes with a Discussion.

## Materials and methods

### Sample

We recruited 139 participants (from an original sample of 144 participants, from which 5 participants did not show up), primarily through the database of the Experimental Laboratory of the School of Economics at the University of Edinburgh—called BLUE (Behavioural Laboratory at the University of Edinburgh), as well as through posters and leaflets on campus. The advertisement and recruitment emails are attached in Appendix A in S1 File.

Participants were required to be at least 18 years old and students at the University of Edinburgh and could not have any pre-existing medical condition. The experiment thus targeted a healthy student population. The study was approved by the School of Economics Ethics Committee at the University of Edinburgh. The slogan used in the advertisements was "Feeling a bit stressed?", targeting students with relatively high levels of anxiety at the start of the study. This was done in order to maximise the chances of inducing an exogenous difference in chronic stress between the treatment and control groups. However, it is likely that such a slogan would attract the attention of many students, as a recent survey by the National Union of Students [39] found that 92 percent of respondents reported feelings of mental distress, including feeling down, stressed and demotivated during their time in higher education. Thus, it is likely that most students at the university "feel a bit stressed".

It is important to point out, however, that, unlike in previous studies, the participants in our experiment have not self-selected into the treatment and are not paying for it, reducing the risk of associated biases. The prospective participants did not know beforehand what the interventions would be.

### Experimental interventions

**Treatment: Mindfulness based stress reduction programme.** The Stress Management Programme consisted of the "Be Mindful Online" mindfulness course. Combining elements of MBSR and Mindfulness Based Cognitive Therapy (MBCT), the course was developed by leading UK mindfulness instructors and is run by the Mental Health Foundation and Wellmind Media. Participants are given an individual login to the course website (http://www.bemindfulonline.co.uk), which provides instructional videos to guide formal meditation. The impact of the course on stress and anxiety has been evaluated by [19].

The course is designed to be taken over four weeks, with a total of 10 interactive online sessions lasting 30 minutes each. The course starts with a three-minute introduction video. This is followed by a questionnaire (including the 10-item version of the Perceived Stress Scale (PSS) of [40]). It also contains the Patient Health Questionnaire (PHQ-9) and the Generalised Anxiety Disorder Assessment (GAD-7). This is followed by an orientation video, which also prompts participants to write down their intentions. During the course, participants are instructed in both formal (including sitting meditation and body scan) and informal (incorporating mindfulness into daily activities) meditation techniques, through videos, assignments, and reminder emails. Participants are asked to practise exercises for both kinds of technique each week between online sessions. Upon completing the course, participants are asked to complete the same questionnaire as in the introduction session of the course.

As participants were asked to follow the programme on their own, we could not enforce compliance. However, the online platform includes a web-based administration system to track participants' activity. In addition, weekly laboratory sessions were held to maintain engagement with the participants and gather self-reported information on their experience of the course (part of the weekly questionnaire, which also included questions about participants'

feelings and health-related behaviours during the previous week). Thus, we are able to study in detail the extent to which participants engage with the programme.

**Control intervention: Historical documentary series.** The control group was asked to watch the documentary series "BBC Ancient Worlds", which was provided to them via email link each week after their visit to the laboratory. This activity was chosen because it would require a similar amount of the participants' time as the MBSR protocol, in order to avoid making the treatment group busier and reducing the time available for health-related activities such as going to gym, etc. Participants in the control group were also asked to come to the laboratory once a week to fill in a questionnaire and provide feedback on the previous week's documentary, allowing us to track their degree of engagement with the programme.

It is plausible to be concerned that watching the BBC ancient world series itself might have a stress-reducing effect. In order to explore the possibility of such effects, we asked the participants to evaluate how useful they found the documentary series for relaxation purposes as part of the weekly feedback. On average, the responses were neutral, indicating slightly lower relaxing effects than reported by the treatment participants for the MBSR intervention (see Appendix B in S1 File for details). Thus, based on these statistics, we do not see evidence indicating that the control intervention would have a stress-reducing effect.

## Experimental procedure

The experimental sessions started in October 2014 and were held at the same time and day every week for each participant, with a total of eight groups each week, spread over three different times on three days. In order to minimise the chance that students would find out about the other intervention, randomisation was conducted at the group level. Table 1 presents a summary of the experimental procedure. Sessions 1, 6 and 7 (pre- and two post-intervention sessions) were longer than the sessions that took place during the intervention. In the the first and sixth sessions we also collected saliva samples to measure cortisol response to a stressful task.

**Table 1. Experimental procedure.**

| Session | Date | Content |
|---|---|---|
| 1 | Week of 20/10/2014 Pre-intervention | 1. Lifestyle and stress survey<br>2. Saliva sample I<br>3. Stressful task<br>4. Decision making tasks<br>5. Saliva sample II<br>6. Further survey questions<br>7. Picture rating task<br>8. Saliva sample III |
| 2 | Week of 27/10/2014 | feedback and short survey |
| 3 | Week of 3/11/2014 | feedback and short survey |
| 4 | Week of 10/11/2014 | feedback and short survey |
| 5 | Week of 17/11/2014 | feedback and short survey |
| 6 | Week of 24/11/2014 Post-intervention | same as in Session 1 |
| 7 | Week of 16/3/2015 5-month follow-up | 1. Lifestyle and stress survey<br>2. Stressful task<br>3. Decision making tasks<br>4. Further survey questions<br>5. Picture rating task |

The structure of Sessions 1 and 6 was as follows. Participants were publicly informed about the structure of the session. They then started the computerized survey, beginning with questions relating to their lifestyle and self-reported stress (including the PSS). When all participants had completed this section, the first sample of saliva was collected simultaneously from all participants in the group. This was followed by the stressful task.

The stressful task involved a combination of testing cognitive ability, time pressure, monetary reward/loss, and social pressure. Full details on the task are provided later. The task was designed to be new to participants in each session in order to avoid participants getting used to it, which could reduce its effectiveness as a stressor. After completing the task and providing feedback on its difficulty and stressfulness, participants proceeded with survey questions on decision-making and decision-making tasks. The second saliva sample was collected precisely 15 minutes after the end of the stressful task, at a time when a peak in cortisol concentrations in response to the stressful event should be expected. Decision-making tasks aimed at eliciting risk and time preferences followed, after which participants answered further background questions (including basic demographic questions in Session 1). The third cortisol sample was taken 23–24 minutes after the second one, by which time the recovery of cortisol levels is expected. In order to provide participants with a neutral activity during the remaining time before the final cortisol sample could be taken, participants were asked to view a series of 30 pictures of humans and 30 pictures of wildlife, rating these respectively on attractiveness and how much they liked the pictures. This task was chosen to fill the time between the two saliva collections in a way that would allow for recovery from the stressful task. Finally, participants were called individually to receive their payments for the session.

Session 7 followed the same procedure as Sessions 1 and 6, but without collection of saliva samples. For Sessions 2–5, participants were asked to complete a short survey asking for feedback about their engagement with the intervention, as well as questions on their health-related behaviours during the previous week.

## Hypotheses and outcome variables

We will now describe the outcome variables we are interested in, as well as our hypotheses regarding the direction in which these variables could be affected by mindfulness training. These include: (1) measures of engagement and compliance with the programmes, (2) measures of chronic stress and response to a stressful situation, (3) measures of behaviour.

**Attrition and compliance.** Our first key outcome of interest is the level of engagement of participants with each of the protocols. The interventions received by both the treatment and the control group require commitment from the participants—in each case the interventions involve watching a video at home and showing up to the laboratory every week. We chose the control intervention so that the degree of commitment required would be similar and, therefore, we do not expect compliance and attrition to systematically differ across treatment and control groups. But one could expect, for example, that certain psychological characteristics such as impulsiveness, impatience and present bias may be correlated with the probability of dropping out. Because we collected a large set of variables at baseline, we are able to test this hypothesis directly.

Our first hypothesis is as follows:

*Hypothesis 1*—Attrition rates will be similar across interventions and positively correlated with psychological characteristics such as impulsiveness, impatience and present bias.

We construct several measures to determine the degree of engagement of participants with the programmes. First, we record participants' attendance at each session. Second, we employ

three different strategies to measure compliance with the programme. One is based on self-reports of engagement in various leisure activities, which are presented in a list format. Meditation is one of the listed activities and participants are asked to report how frequently they have engaged in each activity during the previous week. Another measure is based on summaries participants are asked to write about the contents of the latest lesson (MBSR intervention) or episode (control intervention) in each weekly session. We create an indicator to reflect accuracy of the report (equal to 1 if what they wrote is correct and 0 otherwise). The last measure is based on records of online activity that we obtained from the organisation running the online MBSR course. We have detailed information about the activity and progress of each participant. We use this information to construct a variable indicating how far the participants have progressed with the course.

**Chronic stress and short-run response to a stressful situation.** Because the mindfulness training aims at both decreasing overall anxiety levels and improving the ability to cope with stressful situations, we are interested in measuring both chronic stress levels and the short-run response to a stressful situation (similar to what a student is likely to encounter in her or his daily life).

Self-reported measures of stress are included in the survey questions completed by participants prior to beginning the stressful cognitive task. These measurements are based on the Perceived Stress Scale (PSS), using the 10-item version of the PSS [40]. We extend the PSS with two questions that measure academic stress, which can be particularly relevant among university students. The Perceived Stress Scale (PSS) of [40] is a widely used stress measure, capturing the extent to which an individual perceives events in the previous month as overwhelming or uncontrollable. Several studies of mindfulness interventions have reported reductions in PSS scores (see [19]). In our analysis, we use as an outcome variable the sum of the scores of the 10-item PSS version.

We also collected information on stressful events to which students may have been exposed. Sources of stress are measured with a substantially shortened version of the Adolescent Perceived Events Scale (APES, based on [41]), including a selection of questions most relevant to a student population from the 90-item APES. We use a variable indicating the sum of stressful events the participant faced in the previous month, and test whether her response (in terms of PSS score) differed across treatments. Because mindfulness is supposed to improve coping skills, the hypothesis is that participants in the MBSR treatment should respond less to stressful events.

Following most studies in the literature, we also collect self-reported measures of well-being. The well-being questions were taken from the UK Labour Force Survey. asking respondents the following standard questions: "Overall, how satisfied are you with your life nowadays?"(in weekly surveys: the previous week), which we will refer to as "life satisfaction", and "Overall, how happy are you these days?", which we will refer to as "happiness". We also ask how anxious they feel these days ("anxiety these days") and how anxious they feel right now ("anxiety now"). Participants were asked these questions every week.

**Short-run response to a stressful situation.** The second outcome of interest in relation to stress is the ability to cope with a stressful situation. Participants were asked to perform a task aimed at inducing stress through a combination of testing cognitive ability/knowledge, time pressure, monetary rewards/losses, and social pressure/shame (See [42] for a synthesis of laboratory research on acute stressors.) Because stress responses decline with habituation to a particular stressful situation [43], different stressful tasks were chosen for the pre-and post-intervention sessions.

In the pre-intervention session (Session 1), the task consisted of a computerized cognitive ability and knowledge test, combining numerical, spatial, and verbal reasoning questions with

general knowledge questions. Students were informed that the average student would be expected to be able to answer all questions. Each question was presented on a separate page with a 20 second countdown timer ticking in the top right-hand corner of the page. Students were informed of the requirement of answering 70% of questions correctly in order to participate in a lottery to win one of the two £50 prizes.

In the post-intervention session (Session 6), the task consisted of a cognitive ability and knowledge test that was performed publicly in the laboratory. All participants were asked to stand up in the lab and questions were read aloud by the experimenter, as well as being displayed on a large screen. Immediately after reading a question, the experimenters called upon a randomly selected participant to choose the correct answer to the multiple-choice question. If the given answer was incorrect, participants were informed of this and asked to try another answer. This was repeated until the correct answer had been given. The task consisted of 36 questions. Participants were each endowed with £12 at the beginning of the task, losing £1 for every minute expired on the test. This design was chosen to add social pressure to the task, similar to the Trier Social Stress Test of [44], but with the additional pressure of joint incentive payment.

Finally, in the five-month follow-up session (Session 7), participants were asked to take a computerized Stroop test [45, 46]. Participants were sequentially shown names of four different colours (red, blue, yellow, and green) on the screen, written either in congruent or incongruent colour. They were asked to indicate the colour in which the word was written, by clicking on one of four buttons labelled with the colour names. Upon selecting an answer, the next colour name would immediately appear on the screen. This was repeated 96 times. Participants received one penalty point for each second spent on the task, and one penalty point for each mistake made. They were informed that the two participants with the fewest penalty points would earn a bonus of £50 each.

In each session, directly after completing the task, participants were asked to rate how stressful, difficult, and enjoyable they found the task. This gives us a self-reported measure of the acute stress response. We also asked them to predict their relative performance on the task, before and after having completed it.

In addition, we measured participants' stress response using saliva measurements of cortisol levels, following a standard protocol (we refer to the website of salimetrics for the full description of the protocol). Increased cortisol levels can be measured in saliva between 10 to 20 minutes after exposure to a stressor. If there are no further stressors, cortisol levels should return to their initial level within a short period (between 20 to 40 minutes). This is called the "recovery period". If a person experiences stress for a sustained period of time, she could experience what is called "adrenal fatigue", which leads to low levels of cortisol, a weak response to stressors and a longer recovery period [47].

Saliva samples were collected three times during the experimental session using Salivette collection devices. The saliva samples were analysed by a professional laboratory (Salimetrics). These samples were collected for the initial session and for the post-intervention session, but not for the follow-up session.

Summarising the expected effects on chronic stress and stress response, our second hypothesis is as follows:

*Hypothesis 2*—Participants in the MBSR programme will be better able to cope with stressful situations. As a consequence, chronic stress should decrease and they should be less affected by and recover faster from stressful events.

**Decision-making.**   Finally, we are interested in evaluating how mindfulness affects decision-making. Risk and time preferences presumably play a key role in decisions that are inter-

temporal in nature. We use standard experimental techniques to elicit measures of risk and time preferences.

We use the "Bomb Risk Elicitation Task"(BRET), an intuitive procedure aimed at measuring risk attitudes [48]. Subjects decide how many out of 100 boxes to collect, but are informed that one of the boxes contains a bomb. Earnings increase linearly with the number of boxes collected, but participants receive nothing if the boxes they collect include the one that contains the bomb. Essentially, the task presents 100 lotteries which are described fully in terms of outcomes and probabilities by a single parameter (number of boxes collected). In our experiment, earnings per box are £0.05, i.e., participant earnings are equal to the number of boxes collected divided by 20 (unless the bomb is collected). The major advantage of the BRET, compared with other risk elicitation tasks, is that it requires minimal numeracy skills. The task allows estimation of both risk aversion and risk-seeking, and is not affected by the degree of loss aversion.

[48] present both a static and a dynamic version of the BRET (note that in both versions of the BRET the location of the bomb is only revealed after participants have made their final decision on the total number of boxes they would like to collect). We implement a static version, with participants using a slider to choose how many boxes to collect. In contrast to the dynamic version, in which boxes are collected as time passes and subjects need to decide when to stop collecting boxes, our setup does not introduce any role for time preferences in the decision of how many boxes to collect. Subjects can also revise their decision upward and downward until they are satisfied with their choice. The number of boxes collected is used as the measure of risk aversion. The more risk averse the subject is, the fewer boxes she will collect.

In addition, we construct a non-linear measure of risk aversion, using the approximation of [48]. Assuming a classic power utility function, the coefficient of relative risk aversion (RRA) can be approximated as $1 - \frac{n}{100 - n}$, where n is the number of boxes collected.

How should we expect mindfulness to affect risk-taking behaviour? There is a theory that risk-taking is linked to executive function. For example, there is evidence that risk-taking observed during adolescence may be due to insufficient prefrontal executive function compared to a more rapidly developing subcortical motivation system [49]. Thus, if mindfulness training improves executive function, then we would expect it to decrease risk-taking.

Next, we measure impulsiveness and time preferences using both self-reported and incentivised measures. In order to measure self-reported impulsiveness, we use the Barratt Impulsiveness Scale [50]. This is a widely used measure of impulsiveness, including 30 questions assessing various impulsiveness traits (such as self-control, perseverance, and attention). Each item is reported on a four-point scale, with the total score ranging from 30 (low impulsivity) to 120 (high impulsivity).

For the incentivised measure, we are interested both in eliciting subjects' discount rates and in testing whether their preferences are time-consistent. A simple way to determine time consistency is to offer individuals the choice between smaller amounts of money in the present and larger amounts in the future (i.e., today versus in one week), and then also offer them the identical choice between these rewards shifted further into the future (i.e., four months versus four months and one week). We follow the literature in asking subjects to make such choices for various different monetary rewards. If a subject chooses the smaller reward in the first scenario, but the larger one in the second (so-called static preference reversal), this reveals the subject's present bias. The Tables in Appendix D in S1 File display the choice scenarios for Sessions 1, 6 and 7. Participants were informed in each session that one of their decisions would be randomly selected and implemented at the end of the session. While in Session 1 the monetary rewards were small (ranging from £2 to £4) and everyone received the selected payments,

in Sessions 6 and 7 the rewards were higher (ranging from £30 to £35), but only two randomly selected participants in each session received the payments associated with their decision.

Opting for future payment introduces additional uncertainty and requires subjects to trust the experimenter to pay in the future, introducing variables other than time preference. To keep transaction costs to a minimum, we chose to either provide future payments during pre-scheduled lab-sessions, or give payment via a voucher card, which could be loaded remotely, without the subject having to come to the laboratory. This procedure, combined with the fact that the experimenters are known to use the BLUE lab regularly, should serve to minimize potential trust issues in our participants.

We construct two summary measures of time preferences using these incentivised experiments. First, we count the number of times the participant preferred to receive the money on the day of the session rather than later. We call this variable *impatience*. Second, we construct an indicator of whether the participant exhibits time-inconsistent preferences (*present bias*), preferring to receive a smaller amount of money today over a larger sum at a later date, while preferring the greater and later payment when offered a similar choice between payments on two later dates. We call this binary variable *present bias*. There are three cases of inconsistent choices (i.e., people switching more than once between earlier and later dates), which we exclude from our analysis.

Because mindfulness has been shown to increase executive function, we hypothesize that greater self-control could lead the treatment group to become less present-biased than the control group. Note that, since the core exercises associated with mindfulness involve focusing the mind on the *present*, it is not necessarily obvious that this will be the case. However, there is little evidence pointing in the direction of this opposite effect. Our experiment is, however, the first to consider the effect of mindfulness on a standard measure of present bias.

The hypothesis of the effects of MBSR on the measured behaviours is summarized as follows:

*Hypothesis 3—The MBSR programme will reduce risk-taking, increase patience and reduce present bias.*

We also collected self-reported information on lifestyle behaviour such as smoking, eating, alcohol consumption and sleeping habits of our subjects, and a measure of preferences for "healthy foods", using a revealed preference approach. We refer to Appendix H in S1 File for an analysis of these behaviours.

## Empirical strategy

We estimate the reduced form effect of participating in the MBSR intervention on the outcome variables described above using the following differences-in-differences specifications. Specification (1) is used for outcome measures taken only at the baseline and Sessions 6 and 7, while specification (2) is used for outcome measures that are measured at each session.

$$Y_{it} = \alpha + \beta \ MBSR_i + \gamma_6 \ MBSR_i \times Session6_t + \gamma_7 \ MBSR_i \times Session7_t + \delta_t week_t + \phi X_i + \eta_i + \epsilon_{it} \quad (1)$$

$$Y_{it} = \alpha + \beta \ MBSR_i + \sum_{K=2}^{K=7}(\gamma_K \ MBSR_i \times SessionK_t) + \delta_t week_t + \phi X_i + \eta_i + \epsilon_{it} \quad (2)$$

where $Y_{it}$ is an outcome variable measured for individual $i$ in week $t$. *MBSR* is a dummy variable equal to 1 for individuals in the MBSR Treatment. The *Session* variables are dummy variables that equal 1 if the outcome is measured in that particular session, where *Session*7 corresponds to the five-month follow up session. $X_i$ is a vector of individual characteristics

such as gender, age, ethnicity, a dummy for being an undergraduate student and Body Mass Index at baseline (i.e., week 1). $\eta_i$ is an individual specific random effect and $\epsilon_{it}$ is a white noise error term. We check robustness of our results to the exclusion of the control variables ($X_i$). We also perform the Hausman test, which tests the null hypothesis of orthogonality (no correlation between the regressors and the individual fixed effects $\eta_i$). The test results do not reject the null, implying that our parameter estimates are consistent when estimated using the random effects specification.

Note that attrition can potentially play an important role in the analysis of all outcome variables. Similarly, we cannot be sure that all students have fully complied with the protocol to which they were assigned. So our estimates will always be Intention-to-treat estimates. We first discuss attrition and compliance, and then move to the other outcome variables.

## Results

### Baseline characteristics

We first present baseline characteristics of our sample, as well as information on background socio-economic characteristics we collected in the initial session. We use these baseline characteristics to check for balance in randomisation and, later on, for evaluating the implications of attrition.

Table 2 presents summary statistics for our sample of participants at baseline to evaluate balance across treatment and control samples. In each panel, we report summary statistics (for the pooled sample in Column (1), the treatment sample in Column (2), and the control sample in Column (3)). We test whether the difference is statistically significant in Column (4).

Panel A presents basic individual characteristics that will be used in the analysis as control variables. The average subject in the whole sample is 24.36 years old. About 65 percent of our subjects are female and a similar proportion are white. The average subject weighs about 63.8 kilograms and has a body mass index (BMI) of 21.83. Around 87 percent of our subjects are undergraduate students, while the remaining 13 percent are graduate students.

Panels B and C of Table 2 present summary statistics for the main outcome variables. We start with self-reports of chronic stress, as well as subjective and emotional well-being. In terms of life satisfaction, Panel B shows that the average respondent scores 8.02 on a 11 point Likert scale, with a score of 7.86 in terms of being "happy these days". While students seem to be relatively satisfied with their lives, they still report a high level of anxiety. On an 11 point Likert scale, where 1 represents least anxious and 11 represents most anxious, on average, subjects in our experiments score around 6.7. This highlights that Anxiety appears to be a common problem for our sample of student subjects. Providing a more in-depth measure of stress, we also report participants' Perceived Stress Scale (PSS) score. This is based on 10 questions about the frequency of certain thoughts and feelings associated with stress, each answered on a scale from "Never" to "Very Often" (coded as 0–4, with 0 representing "Never" and 4 representing "Very Often"). Thus the highest possible PSS score would be 40. In our baseline sample, the average PSS score is 17.78. This is comparable to PSS scores in similar samples in previous studies. For example, based on samples of university students in the US, [51] report a mean value of 19.56 and [52] report a mean of 18.3 on the ten-item PSS. Our average score is lower than the mean scores of 23.04 and 22.4 reported by [14, 19], respectively, based on samples of individuals choosing to complete an online mindfulness course.

Panel C reports the behavioural measures at baseline, both those based on self-reports (impulsivity and patience), and those based on incentivised revealed-preference measures (present bias and risk aversion). Only a small proportion of participants is present-biased (8%) at baseline, which is surprisingly low. The average value of impulsivity is in the lower half of

**Table 2. Baseline characteristics.**

| Baseline characteristics | | | | | | | |
|---|---|---|---|---|---|---|---|
| | [1] | | [2] | | [3] | | [4] |
| | Total | | Treatment | | Control | | Diff |
| **Variables** | **Mean** | **SD** | **Mean** | **SD** | **Mean** | **SD** | **Mean** |
| *Panel A: Individual Characteristics* | | | | | | | |
| Age | 24.36 | 3.61 | 23.76 | 1.92 | 24.92 | 4.60 | 1.16* |
| Female | 0.65 | 0.48 | 0.69 | 0.47 | 0.61 | 0.49 | -0.08 |
| White | 0.65 | 0.48 | 0.66 | 0.48 | 0.64 | 0.48 | -0.02 |
| Weight (kg) | 63.81 | 10.16 | 64.09 | 10.57 | 63.56 | 9.83 | -0.53 |
| Body mass index (BMI) | 21.83 | 2.59 | 22.25 | 2.73 | 21.44 | 2.41 | -0.81 |
| Undergraduate | 0.87 | 0.34 | 0.90 | 0.31 | 0.85 | 0.36 | -0.05 |
| *Panel B: Stress and Wellbeing* | | | | | | | |
| Perceived stress score (scale: 0–40) | 17.78 | 6.00 | 18.49 | 5.81 | 17.11 | 6.14 | -1.38 |
| Anxious these days (scale: 1–11) | 6.76 | 2.42 | 7.10 | 2.43 | 6.43 | 2.39 | -0.67 |
| Anxious now (scale: 1–11) | 5.50 | 2.43 | 6.01 | 2.45 | 5.03 | 2.33 | -0.99 |
| Life satisfaction nowadays (scale: 1–11) | 8.02 | 1.47 | 8.01 | 1.32 | 8.03 | 1.60 | 0.01 |
| Happiness these days (scale: 1–11) | 7.86 | 1.62 | 7.78 | 1.60 | 7.93 | 1.65 | 0.15 |
| Happiness now (scale: 1–11) | 7.40 | 1.61 | 7.46 | 1.44 | 7.35 | 1.77 | -0.12 |
| Things worthwhile (scale: 1–11) | 8.22 | 1.61 | 8.00 | 1.70 | 8.42 | 1.51 | 0.42 |
| *Panel C: Behavioural measures* | | | | | | | |
| Present bias (0/1) | 0.08 | 0.27 | 0.07 | 0.26 | 0.08 | 0.28 | 0.01 |
| BIS total score (30 to 120) | 64.34 | 9.47 | 65.01 | 10.25 | 63.71 | 8.70 | -1.31 |
| Risk aversion (BRET) | 45.65 | 20.19 | 48.01 | 22.36 | 43.44 | 17.81 | -4.57 |
| Impatience (0 to 10) | 0.48 | 1.10 | 0.42 | 1.03 | 0.54 | 1.16 | 0.12 |
| Observations | 139 | | 67 | | 72 | | |

Notes:

***p < 0.01,

**p < 0.05,

* p < 0.1.

the impulsivity scale and is comparable to other recent studies involving students in the UK (see [53]). In terms of patience, the majority of the students prefer all of the later options to the earlier ones, thus the indicator of impatience at Session 1 is on average very low. Note, that impatience becomes more common in Sessions 6 and 7, when the monetary rewards are higher. Finally, the mean value of our measure of risk aversion, corresponding to the number of boxes collected in the BRET, is 46, which is very close to the mean observed in Crosetto and Filippin (2013).

## Attrition and compliance

We begin by testing *Hypothesis 1*. Both interventions require some degree of commitment from the participants. Our data allow us to study the determinants of continued participation in the study and, in particular, engagement with the mindfulness protocol. One would expect that certain behavioural characteristics such as impulsiveness, impatience and present bias may be correlated with the likelihood of attrition. Because we have collected a large set of variables at baseline, we are able to test this hypothesis directly.

We start with the information on attendance. The number of subjects in both the treatment and control groups declined over time due to attrition. In Session 6, 17 of the 67 original subjects in the treatment group (representing 25%) and 11 of the original 72 subjects in the control group (representing 15.3%) did not attend the experimental session. The non-attendance rate in Session 7 was 41.8% in the treatment group and 29.2% in the control group. Concern about bias in the estimation results due to attrition thus seems justified.

First, we analyse the determinants of attrition using attrition probits [54]. Attrition probits consist of estimates of binary-choice models for the determinants of attrition in later periods as a function of base year characteristics. We estimate separate attrition probit models for the treatment and control groups. We include a rich set of baseline characteristics in the models, but have to exclude some variables to avoid strong multicollinearity (anxiety now, happiness now) and perfect prediction (present bias). We will come back to the latter, since it is a variable we thought could be correlated with engagement. The dependent variable is a binary indicator of being present at Session 6 or 7.

The results presented in Table 3 show that, although there are some significant coefficients in the attrition probit models, there is no systematic relation between the baseline characteristics and attrition. The personal characteristics that are significantly related to attrition are those characteristics for which we control in our estimations. We also see that anxiety these days significantly reduces the probability of remaining in the sample within the treatment group. If the MBSR program is more effective among the subjects who report anxiety, then this selectivity can lead to underestimation of the beneficial effect of the program on anxiety. Six individuals in the control group, coded as present-biased in Session 1, have to be excluded due to perfect prediction of non-attrition by present bias. To gauge the effects of present bias on attrition, we tested for a simple mean difference in present bias as measured in Session 1

**Table 3. Attrition probits (marginal effects on non-attrition).**

| | Treatment Session 6 | | Control Session 6 | | Treatment Session 7 | | Control Session 7 | |
|---|---|---|---|---|---|---|---|---|
| | Marginal effect | SE | Marginal effect | SE | Marginal effect | SE | Marginal effect | SE |
| **Personal characteristics** | | | | | | | | |
| Age | -0.041 | 0.025 | 0.011* | 0.007 | -0.047 | 0.031 | 0.036** | 0.014 |
| Female | 0.095 | 0.104 | -0.035 | 0.065 | 0.408*** | 0.132 | -0.069 | 0.113 |
| White | -0.126 | 0.091 | 0.209* | 0.125 | -0.101 | 0.146 | 0.336** | 0.151 |
| Body mass index (BMI) | 0.036* | 0.021 | -0.014 | 0.011 | 0.046 | 0.030 | 0.007 | 0.023 |
| **Stress and subjective well-being** | | | | | | | | |
| PSS | 0.010 | 0.013 | 0.002 | 0.007 | 0.009 | 0.016 | -0.013 | 0.011 |
| Anxious these days | -0.060* | 0.034 | 0.039** | 0.018 | -0.096** | 0.044 | 0.026 | 0.034 |
| Anxious now | -0.004 | 0.027 | -0.052*** | 0.020 | -0.005 | 0.037 | -0.007 | 0.032 |
| Life satisfaction nowadays | 0.091** | 0.046 | -0.036 | 0.029 | 0.029 | 0.061 | -0.073 | 0.058 |
| Happiness these days | -0.011 | 0.042 | 0.031 | 0.026 | -0.023 | 0.054 | -0.038 | 0.051 |
| **Behavioural measures** | | | | | | | | |
| Impulsivity (BIS) | -0.003 | 0.004 | -0.009** | 0.005 | -0.003 | 0.007 | -0.005 | 0.008 |
| Risk aversion (BRET) | 0.001 | 0.002 | 0.002 | 0.002 | 0.002 | 0.004 | 0.003 | 0.003 |
| Impatience | -0.052 | 0.056 | 0.134* | 0.069 | -0.007 | 0.079 | 0.073 | 0.067 |
| Present bias | -0.341 | 0.365 | | | -0.211 | 0.350 | -0.112 | 0.263 |
| No. of individuals | 67 | | 65 | | 67 | | 71 | |

*, **, *** indicate significance levels at 10%, 5% and 1% respectively

**Table 4. Comparison of baseline means of the non-attrited subsamples of treatment and control groups.**

| | Present at Session 6, treatment-control | | Present at Session 7, treatment-control | |
|---|---|---|---|---|
| | **Diff.** | **SE** | **Diff.** | **SE** |
| **Personal characteristics** | | | | |
| Age | -1.478** | 0.732 | -1.991** | 0.874 |
| Female | 0.097 | 0.09 | 0.252** | 0.097 |
| White | -0.032 | 0.091 | -0.045 | 0.101 |
| BMI | 1.269** | 0.498 | -1.356** | 0.579 |
| **Stress and subjective well-being** | | | | |
| PSS | 0.992 | 1.165 | 1.148 | 1.35 |
| Anxious these days | 0.192 | 0.456 | -0.062 | 0.533 |
| Anxious now | 0.724 | 0.458 | 0.297 | 0.525 |
| Life satisfaction nowadays | 0.233 | 0.273 | 0.299 | 0.312 |
| Happiness these days | -0.034 | 0.319 | 0.056 | 0.370 |
| **Behavioural measures** | | | | |
| Impulsivity (BIS) | 1.412 | 1.881 | 1.090 | 2.033 |
| Risk aversion (BRET) | 3.081 | 3.831 | 2.428 | 4.004 |
| Impatience | -0.343 | 0.213 | -0.268 | 0.241 |
| Present bias | -0.058 | 0.049 | -0.047 | 0.057 |

*, **, *** indicate significance levels at 10%, 5% and 1% respectively

between the original sample and the sample present in Sessions 6 and 7. We found no significant differences.

As a second test of attrition, we look at whether the treatment and control samples that are present in Sessions 6 and 7 are still comparable in terms of their baseline characteristics. This check can reveal whether there is asymmetric attrition between the treatment and the control groups (on observable characteristics). We test for equality of the same set of baseline characteristics that we used in the attrition probit models. The results are presented in Table 4. There are statistically significant differences in age, gender and BMI between the treated and control individuals at the baseline, but these are relatively small. These are also characteristics that we control for in the empirical specifications. More importantly, we do not see significant differences in terms of risk attitudes, patience or impulsiveness, our key outcome variables of interest.

Thus, our hypothesis that attrition would be positively related with impulsiveness, impatience and present bias (Hypothesis 1) is not supported by the data. Also, we do not see that engagement with the protocols is correlated with psychological traits or behavioural measures such as impulsiveness and impatience. This supports that mindfulness can possibly be an efficient instrument to target subjects with various behavioural characteristics. These findings also reduce concerns about the analysis of behavioural measures suffering from bias due to attrition. We provide a further check of the importance of attrition in Appendix F in S1 File, where we re-estimate the results on PSS and anxiety measures using the non-attriting sub-sample.

The next variables of interest are the degree of engagement and compliance with the interventions. We have designed three strategies to measure these. First, we asked participants to report every week to what extent they engaged in various activities to relax, such as meeting with friends, going to the theatre, etc. (see Appendix C in S1 File for full questionnaire). Meditating is one of the activities they were asked about. Fig 1 shows the average report on the extent to which participants meditate, with 0 being never or less than once a week and 3 being

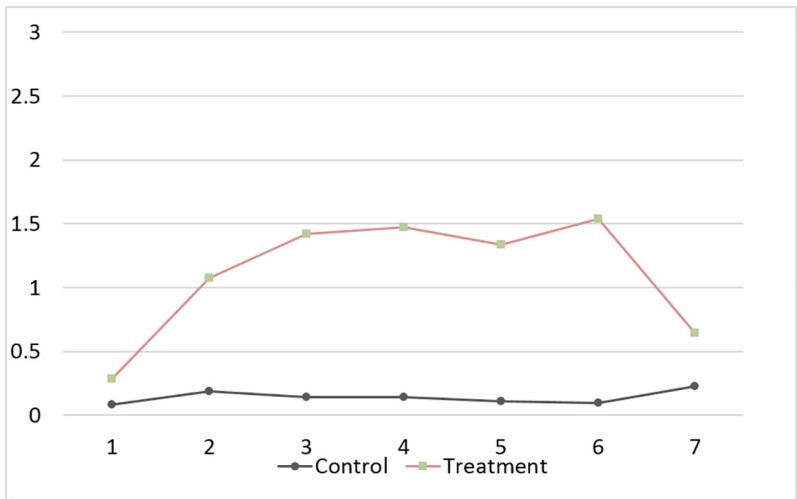

**Fig 1. Compliance.** Average frequency of meditation (0-less than once a week or never to 3-almost every day), by session and treatment.

almost every day. We report the difference-in-difference analysis in Appendix E in S1 File. We find a significantly positive treatment effect on the frequency of mediation during Sessions 2–6. The effect remains positive but becomes statistically insignificant during the follow-up session five months later.

Second, we asked participants to describe the contents of the weekly mindfulness lesson (in at least 100 words). We also asked the participants in the neutral intervention to describe the documentary episode they were asked to watch. We constructed a dummy variable indicating whether the text was indeed an accurate description of the lesson/episode. We coded the response as a zero if the description was generic and did not demonstrate that they engaged with the intervention or if they mention not having done the activity at all. Participants complied to a large extent with the interventions. Based on this binary indicator of compliance, more than 90% of the subjects in the control group followed the control intervention every week. Compliance also exceeded 90% in the treatment group every week, except for the first week, when 82% of the subjects followed the MBSR program (this statistic is based on the survey during Session 2). Of course it is worth noting that the number of participants falls over time, and attriting participants are unlikely to still be engaged with the intervention.

Finally, the last strategy to check for engagement and compliance with the interventions involves using data from the website, which tracked participants' activities during the online sessions. The website tracks when participants logged in and completed the various stages of the intervention. By Session 6, 36% of the non-attrited individuals in the treatment group had fully completed the online mindfulness course (while 72% had reached at least Week 4 of the course by this point). By Session 7, the completion rate increased to 59%. The estimated marginal effects on the probability of completing the course (based on probit models) are reported in Table 5. According to these results, indicators of stress, behavioural preferences and health behaviours are not systematically related to the probability of completing the mindfulness course. Therefore, it does not seem likely that systematic selection out of the completion of the course would undermine the effectiveness of the mindfulness program, or our estimation results on the effects of the MBSR would be driven by selection into completing the course.

**Table 5. Probability of completing the course.**

|  | Marginal | |
| --- | :---: | :---: |
|  | effect | SE |
| **Personal characteristics** |  |  |
| Age | 0.086* | 0.046 |
| Female | 0.244 | 0.148 |
| White | -0.051 | 0.140 |
| Body mass index (BMI) | 0.035 | 0.026 |
| **Stress and subjective well-being** |  |  |
| PSS | 0.007 | 0.014 |
| Anxious these days | -0.068* | 0.039 |
| Anxious now | 0.036 | 0.035 |
| Life satisfaction nowadays | 0.005 | 0.060 |
| Happiness these days | 0.007 | 0.053 |
| **Behavioural measures** |  |  |
| Impulsivity (BIS) | -0.001 | 0.006 |
| Risk aversion (BRET) | 0.000 | 0.003 |
| Impatience | -0.034 | 0.090 |
| Present bias | -0.073 | 0.346 |
| No. of individuals | 65 |  |

*, **, *** indicate significance levels at 10%, 5% and 1% respectively

## Effects on chronic stress and stress response

We now turn to *Hypothesis 2*, which relates to the impact of the MBSR intervention on chronic stress and on the response to a stressful situation. These outcomes are the primary targets of the programme.

We use three different sources to construct a measure of chronic stress. The first is based on the total score of the Perceived Stress Scale (measured in the initial session and in the two post-intervention sessions). The second and third are based on responses to weekly questions about how anxious the participants feel "now" and "these days", both on a scale from 1–11. Table 6 reports the treatment effects of the intervention on these three measures from a difference-in-differences estimator (MBSR & Session 6, and MBSR & Session 7 show the post-intervention estimates of the treatment effect).

Results show that the MBSR intervention leads to a significant decrease in participants' PSS scores both in the week immediately following completion of the course (session 6) and at the 5 month follow-up (session 7). The fall is on the order of 10 percent (compared to the baseline average PSS score). The estimated treatment effect resulted from both decreasing levels of stress among the treatment group and increasing levels of stress among the control group. The effect is comparable to the effect found by [14] after 8 and 12 weeks of an internet-based mindfulness program, but smaller than the effect found by [19]. [19] estimate that the online mindfulness course reduces the average PSS score by around 8 points and by a further 1.5 points a month later; however, these estimates are based on a sample of self-selected individuals, without the inclusion of a control group in their analysis. The MBSR intervention also appears to reduce reported anxiety throughout session 2 to 7, but these estimates are mostly not statistically significant. The results indicate that the treatment is more effective in reducing the current level of anxiety (anxiety "now") than the general level of anxiety (anxiety "these days").

**Table 6. The impact of MBSR on Perceived Stress Score (PSS) and anxiety measures.**

| | [1] | | [2] | | [3] | |
| | PSS | | Anxiety Now | | Anxiety These Days | |
| | Coeff. | SE | Coeff. | SE | Coeff. | SE |
|---|---|---|---|---|---|---|
| MBSR | 1.463 | 0.981 | 1.256*** | 0.407 | 0.878** | 0.408 |
| Session 2 | . | . | 0.531 | 0.346 | 0.273 | 0.336 |
| Session 3 | . | . | 0.918*** | 0.327 | 0.302 | 0.275 |
| Session 4 | . | . | 0.227 | 0.371 | -0.095 | 0.304 |
| Session 5 | . | . | 0.344 | 0.351 | -0.141 | 0.306 |
| Session 6 | 0.999* | 0.511 | 0.279 | 0.344 | -0.131 | 0.245 |
| Session 7 | 2.205*** | 0.855 | 0.747* | 0.387 | 0.344 | 0.355 |
| MBSR & Session 2 | . | . | -0.857* | 0.505 | -0.454 | 0.450 |
| MBSR & Session 3 | . | . | -1.163** | 0.520 | -0.882* | 0.458 |
| MBSR & Session 4 | . | . | -0.402 | 0.529 | -0.387 | 0.445 |
| MBSR & Session 5 | . | . | -0.360 | 0.511 | -0.296 | 0.495 |
| MBSR & Session 6 | -1.809* | 0.926 | -0.068 | 0.542 | -0.069 | 0.454 |
| MBSR & Session 7 | -2.464* | 1.320 | -1.095* | 0.650 | -0.765 | 0.582 |
| Intercept | 17.363** | 7.063 | 6.982*** | 2.009 | 8.575*** | 2.116 |
| Individual random effects | Yes | | Yes | | Yes | |
| Control variables | Yes | | Yes | | Yes | |
| No. of individuals | 138 | | 138 | | 138 | |

*Notes*: Robust standard errors;

*** $p < 0.01$,

** $p < 0.05$,

* $p < 0.1$.

We do not find any significant treatment effects on other measures of subjective well-being, including measures of life satisfaction, happiness and "considering things worthwhile".

We conducted a series of specification checks to investigate further the the impact of the MBSR intervention on PSS score and anxiety. First, to check for the importance of attrition, we re-estimated the treatment effects using the sub-sample of individuals who were present at Session 6 or 7. Although the precision of the estimated treatment effects declines, the main conclusions remain robust. These results are reported in Appendix F in S1 File. Next, we estimated the effect of MBSR on the sum of the two indicators of academic stress (worries about grades in the current semester and in the future). We find no significant treatment effects. Finally, while we see that stressful events (measured by the modified Adolescent Perceived Events Scale, APES) increase the PSS score, we do not see evidence that the PSS score of the treatment group responds less to such stressful events.

Next, we examine how the intervention affected the response to a stressful situation. Appendix G in S1 File summarises how stressful, not enjoyable and difficult the participants found each task. We also present an indicator of over-confidence. The binary indicator capturing over-confidence equals one if, before the task, a participant thinks she would perform among the best three or best six people in the room, but, after the task, she does not think she performed among the best three or six. Based on these indicators, while the stressfulness of all three tasks was rated around 6–7 on average on a 10-point scale, the computerised ability and knowledge test was considered on average less enjoyable and more difficult than the post-intervention tasks. Over-confidence was also more prevalent in the first session. Apart from

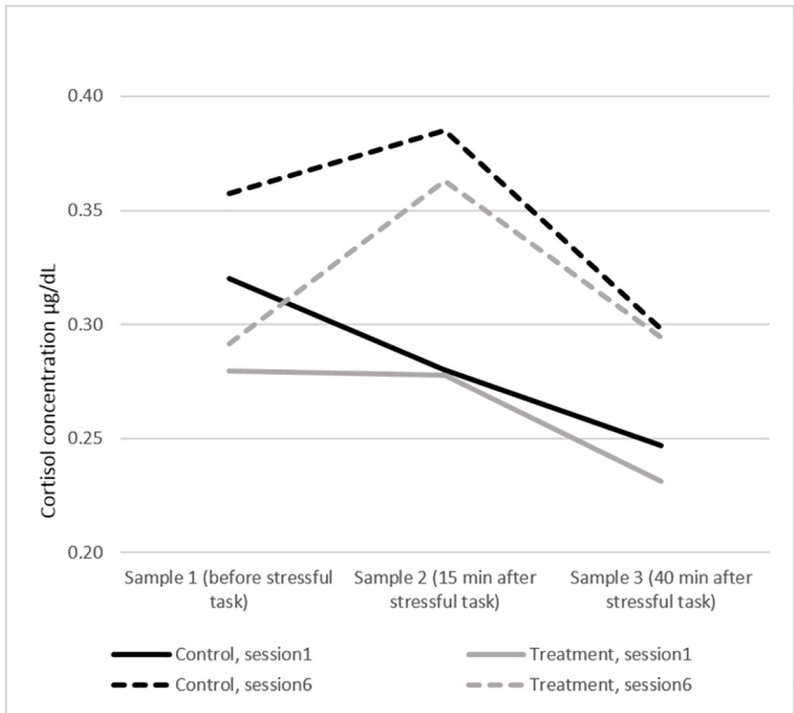

**Fig 2. Cortisol.** Salivary cortisol concentration averages by session and by treatment.

over-confidence in Session 7, there was no statistically significant difference between the treatment and control group with respect to the evaluation of the stressful tasks. In the final session, over-confidence was 14.5 percentage points more prevalent within the treatment group than the control group.

Considering the salivary cortisol measurements, we do not find evidence that the MBSR intervention significantly affected the objective measures of stress levels and stress responses. The average levels of the three cortisol measurements by session and by treatment are displayed in Fig 2. These cortisol levels are within the normal ranges of cortisol concentration.

Summarising, the evidence based on self-reported measures is supportive of *Hypothesis 2*, but the evidence based on physiological measurements is inconclusive.

### Effects on decision-making

We now turn to investigating the effects of the intervention on decision-making, specifically on risk taking and time preferences (*Hypothesis 3*).

The impulsivity and risk measures are identical across Sessions 1, 6 and 7. Table 7 presents difference-in-differences regression results on the impact of our MBSR intervention on risk preferences. Results in Column [1] show that the number of boxes collected decreases in the treatment group relative to the control group (which is indicative of a decrease in risk-taking), but only in Session 7, and the effect is not statistically significant at conventional levels. Using the approximated coefficient of relative risk aversion (Column [2]), the estimated treatment effects are more robust and statistically significant in Session 7. As the coefficient "MBSR" indicates, subjects in the treatment group were initially significantly less risk-averse than those in the control group, as measured by the coefficient of relative risk aversion. The gap between the

**Table 7. The impact of MBSR on risk aversion.**

| | [1] | | [2] | |
| | Risk aversion (BRET) | | Risk aversion (BRET) | |
| | Number of boxes | | RRA coefficient | |
| | Coeff. | SE | Coeff. | SE |
|---|---|---|---|---|
| MBSR | 5.04 | 3.453 | -0.733** | 0.362 |
| Session 6 | -4.081 | 2.693 | 0.138 | 0.111 |
| Session 7 | 2.844 | 2.87 | -0.047 | 0.096 |
| MBSR Session 6 | 0.407 | 4.378 | 0.431 | 0.414 |
| MBSR Session 7 | -4.522 | 3.987 | 0.779** | 0.398 |
| Intercept | 63.203*** | 13.99 | -1.560** | 0.757 |
| Individual RE | Yes | | Yes | |
| Control variables | Yes | | Yes | |
| No. of individuals | 138 | | 138 | |

*, **, *** indicate significance levels at 10%, 5% and 1% respectively

two groups appears to be eliminated by the MBSR treatment. This effect is in line with *Hypothesis 3*.

We next turn to the measures of time preferences. Because the measure of time preferences was not identical across sessions, we conduct a simple difference analysis on patience and present-bias between treatment and control groups for Sessions 1, 6 and 7 separately and report the results in Table 8. For patience, we find that participants in the treatment group became more patient after the intervention, but the effects are not statistically significant. The point estimates and standard errors are quite large, however, so there is a possible issue of statistical power. For present bias, we find that participants have a similar propensity of being present-biased in Session 1, but the treatment group appears less present-biased immediately after the intervention, although the effects are again not statistically significant. We find no significant difference in Session 7 either. However, it is useful to point out that the baseline measure of present-bias was very low (with only 8% of the participants categorised as present-biased). Overall, we take our results as somewhat indicative that patience may have increased and the propensity to be present-biased decreased, but these results are not statistically significant at conventional levels. Thus, we do not find strong support for *Hypothesis 3*.

**Table 8. The impact of MBSR on time preferences.**

| | Impatience | | | Present bias | | |
| | Session 1 | Session 6 | Session 7 | Session 1 | Session 6 | Session 7 |
|---|---|---|---|---|---|---|
| MBSR | -0.018 | -0.179 | -0.120 | 0.016 | -0.113 | 0.026 |
| | (0.178) | (0.411) | (0.486) | (0.033) | (0.101) | (0.121) |
| Intercept | 3.002** | 5.184 | -3.571 | | | |
| | (1.294) | (1.891) | (2.501) | | | |
| Control variables | yes | yes | yes | yes | yes | yes |
| Individual RE | No | No | No | No | Yes | Yes |
| No of individuals | 136 | 112 | 80 | 136 | 112 | 80 |

Notes: Estimates for impatience are OLS, estimates for Present bias are probit (marginal effects reported).

*, **, *** indicate significance levels at 10%, 5% and 1% respectively

**Table 9. The impact of MBSR on impulsivity.**

|  | Coeff. | SE |
|---|---|---|
| MBSR | 0.809 | 1.589 |
| Session 6 | -0.11 | 0.735 |
| Session 7 | 0.217 | 0.756 |
| MBSR Session 6 | 2.229* | 1.174 |
| MBSR Session 7 | 0.983 | 1.298 |
| Intercept | 59.199*** | 9.135 |
| Individual RE | Yes | |
| Control variables | Yes | |
| No. of individuals | 138 | |

Notes:

*, **, *** indicate significance levels at 10%, 5% and 1% respectively.

We should however point out that the results for decision-making tasks are imprecisely estimates (the standard errors are relatively large), pointing to a possible issue of statistical power. It could be that our sample is too small to identify effects. We could not and did not perform ex ante power calculations for all variables since we did not have prior information about their distribution. Prior information was available for the risk taking task from [55]. In their static version of the game, they report a mean of 46.5 boxes collected and a standard deviation of 15.1. With this mean and standard deviation, we were powered to detect an effect size of 15% (increase from 46 to 53 boxes) with power 0.80 and significance at 5%. While such effect may seem large, it is not an unreasonable size effect to expect.

Finally, we look at how mindfulness affects the scores on the Barratt Impulsiveness Scale (BIS). We conduct a difference-in-differences analysis and report the results in Table 9. Here, we document a significant effect in Session 6, but we find that participants in the treatment group increased their average score on the BIS relative to the participants in the control group; that is, if anything, mindfulness appears to have increased impulsiveness rather than decreased it. These survey-based results go in the opposite direction to those we found for patience and present-bias using revealed preference methods. This result seems to be driven by the sub-category of questions related to "Self-control"and, more specifically, four of the 30 questions that make up the Barratt Impulsiveness Scale. In Session 6, participants in the treatment group are statistically significantly more likely to describe themselves as "I am happy-go-lucky"(Barratt item 4) and significantly less likely to say "I am self-controlled"(item 8), "I am a careful thinker"(item 12), or "I am a steady thinker"(item 20). The effect on "I am happy-go-lucky"(Barratt item 4) persists to Session 7 five months later. Given that these are self-reported measures, it is of course possible that doing the mindfulness course has simply made participants think of themselves differently (and possibly more critically, in terms of self-control).

We also collected self-reported measures of lifestyle and an incentivised measure of health behaviour (food choice). We report the results in Appendix H in S1 File. Overall, participants in the MBSR programme appear to adopt somewhat healthier eating habits, but the effects are only significant for one variable, which is a measure of self-reported emotional eating.

## Discussion

As mentioned in the introduction, the question of whether behavioural attitudes are malleable and can be "altered" or "trained" is a relatively new question in Economics. We contribute to

the literature by using a randomized controlled experiment to identify the impact of a mindfulness training programme on behavioural traits and anomalies that play a key role in intertemporal choices. Importantly, our participants did not self-select into the programmes.

Analysing the patterns of attrition and compliance, we find no evidence that behavioural characteristics would predict engagement with the mindfulness program. We find that the mindfulness intervention significantly reduces perceived stress, but the evidence based on physiological measures of stress (cortisol) is less conclusive. There is some evidence that participants may have become more risk averse, but only weak evidence that they become more patient and less present-biased. On the other hand, participants in the MBSR treatment score higher on the Barratt Impulsiveness Scale based on survey questions.

Overall, we conclude that such interventions appear to be effective at targeting people with various behavioural characteristics and reducing "feelings of stress", but the effects on decision-making are unclear. Looking at the set of point estimates we have, we cannot reject the hypothesis that mindfulness may have in fact increased patience and risk aversion, and reduced present-bias, by a significant magnitude. Further research is needed to obtain more robust evidence of the effects of such techniques on decision-making.

## Supporting information

**S1 File. Supplementary materials.**
(PDF)

## Acknowledgments

We thank participants of the Winter Workshop on Labour Economics at the University of Amsterdam, seminar audiences at CREED (University of Amsterdam), CESS (University of Oxford), NYUAD, Innsbruck and University of Gothenburg for constructive comments. All errors and omissions are ours.

## Author Contributions

**Conceptualization:** Yonas Alem, Hannah Behrendt, Michèle Belot, Anikó Bíró.

**Data curation:** Hannah Behrendt, Anikó Bíró.

**Formal analysis:** Yonas Alem, Hannah Behrendt, Michèle Belot, Anikó Bíró.

**Funding acquisition:** Yonas Alem, Michèle Belot.

**Investigation:** Yonas Alem, Hannah Behrendt, Michèle Belot, Anikó Bíró.

**Methodology:** Hannah Behrendt, Michèle Belot, Anikó Bíró.

**Project administration:** Michèle Belot, Anikó Bíró.

**Resources:** Hannah Behrendt, Michèle Belot, Anikó Bíró.

**Software:** Hannah Behrendt.

**Supervision:** Michèle Belot.

**Writing – original draft:** Yonas Alem, Hannah Behrendt, Michèle Belot, Anikó Bíró.

**Writing – review & editing:** Yonas Alem, Hannah Behrendt, Michèle Belot, Anikó Bíró.

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
