## [Decision Letter · Decision Letter 0]

30 Jul 2021

PONE-D-21-20053

Mind Training, Stress and Behaviour - a Randomised Experiment

PLOS ONE

Dear Dr. Belot,

Thank you for submitting your manuscript to PLOS ONE. After careful consideration, we feel that it has merit but does not fully meet PLOS ONE’s publication criteria as it currently stands. Therefore, we invite you to submit a revised version of the manuscript that addresses the points raised during the review process.

As you will see both reports are fairly positive. The only concern is whether the lack of results is due to the sample size. I feel that the author need to discuss the impact of the sample size (or the necessary sample size to have power) on the results. I am not asking for more data.

We look forward to receiving your revised manuscript.

Kind regards,

Pablo Brañas-Garza, PhD Economics

Academic Editor

PLOS ONE

Journal Requirements:

Reviewers' comments:

Reviewer's Responses to Questions

**Comments to the Author**

1. Is the manuscript technically sound, and do the data support the conclusions?

Reviewer #1: Yes

Reviewer #2: Yes

2. Has the statistical analysis been performed appropriately and rigorously? 

Reviewer #1: Yes

Reviewer #2: Yes

3. Have the authors made all data underlying the findings in their manuscript fully available?

Reviewer #1: Yes

Reviewer #2: Yes

4. Is the manuscript presented in an intelligible fashion and written in standard English?

Reviewer #1: Yes

Reviewer #2: Yes

5. Review Comments to the Author

Reviewer #1: This is an interesting study. I like it that the authors have measured stress in different ways and thus replicated prior results with a non-self selected sample.

My only concern is whether the lack of results in the decision-amking tasks is related to the sample size. I would like the authors to discuss what effect sizes their sample are powered to detect.

Reviewer #2: I liked the paper very much. Its clear and very well executed. As authors know they have a very reduced sample but, on the other hand this is a novel idea.

- Table 2: I feel that is better to say "risk taking" rather than "# boxes collected (BRET)"  Compare Table 2 and 4

- perhaps the number of table might be reduced (some of the might be merged)

Additional references to consider:

- On time preferences: https://mpra.ub.uni-muenchen.de/103660/

6. PLOS authors have the option to publish the peer review history of their article (what does this mean?). If published, this will include your full peer review and any attached files.

Reviewer #1: No

Reviewer #2: No

---

## [Author Response · Author response to Decision Letter 0]

3 Sep 2021

Below we provide an answer to the queries. To facilitate reading, we copied and pasted the original comments by reviewers. 

We are very grateful for the comments that have helped improving the paper.

Comments to the Author 1. Is the manuscript technically sound, and do the data support the conclusions? The manuscript must describe a technically sound piece of scientific research with data that supports the conclusions. Experiments must have been conducted rigorously, with appropriate controls, replication, and sample sizes. The conclusions must be drawn appropriately based on the data presented.

Reviewer #1: Yes

Reviewer #2: Yes

2. Has the statistical analysis been performed appropriately and rigorously?

Reviewer #1: Yes

Reviewer #2: Yes

3. Have the authors made all data underlying the findings in their manuscript fully available? The PLOS Data policy requires authors to make all data underlying the findings described in their manuscript fully available without restriction, with rare exception (please refer to the Data Availability Statement in the manuscript PDF file). The data should be provided as part of the manuscript or its supporting information, or deposited to a public repository. For example, in addition to summary statistics, the data points behind means, medians and variance measures should be available. If there are restrictions on publicly sharing data—e.g. participant privacy or use of data from a third party—those must be specified.

Reviewer #1: Yes

Reviewer #2: Yes

4. Is the manuscript presented in an intelligible fashion and written in standard English? PLOS ONE does not copyedit accepted manuscripts, so the language in submitted articles must be clear, correct, and unambiguous. Any typographical or grammatical errors should be corrected at revision, so please note any specific errors here.

Reviewer #1: Yes

Reviewer #2: Yes

5. Review Comments to the Author Please use the space provided to explain your answers to the questions above. You may also include additional comments for the author, including concerns about dual publication, research ethics, or publication ethics. (Please upload your review as an attachment if it exceeds 20,000 characters)

Reviewer #1: This is an interesting study. I like it that the authors have measured stress in different ways and thus replicated prior results with a non-self selected sample. My only concern is whether the lack of results in the decision-making tasks is related to the sample size. I would like the authors to discuss what effect sizes their sample are powered to detect.

Response from the authors: Thank you for your positive view on the paper. Your point about lack of power is well taken. Power calculations were not common at the time we conducted the experiment, and our understanding is that ex-post power calculations are problematic (see for example this discussion on a World Bank Blog:https://blogs.worldbank.org/impactevaluations/why-ex-post-power-using-estimated-effect-sizes-bad-ex-post-mde-not). In our view, it may make sense to perform power calculations for analyses of outcomes that have been previously studied in the literature. For risk, we can base our calculations on the study of Crosetto & Filippin (2013). In their static version of the game, they report a mean of 46.5 boxes collected and a standard deviation of 15.1. With this mean and standard deviation, we were powered to detect an effect size of 15% (going from 46 to 53 boxes) with power 0.80 and significance at 5%.

For the estimates of impatience and present bias, it is harder to find studies that match exactly out outcome variables (other studies have different incentives, so we cannot assume that they provide good estimates of the fraction of present biased individuals, and the discount rates we estimate are also specific to the timelines we used.) So for these variables, we would be reluctant to present ex-post power calculations. We have chosen to acknowledge more explicitly the possibility that the lack of significant results may be due to small sample size.

Reviewer #2: I liked the paper very much. Its clear and very well executed. As authors know they have a very reduced sample but, on the other hand this is a novel idea. - Table 2: I feel that is better to say "risk taking" rather than "# boxes collected (BRET)"  Compare Table 2 and 4

- perhaps the number of table might be reduced (some of the might be merged)

Response from authors: Response from authors: Thank you your positive assessment! We agree and we have implemented your suggestions. We have merged Tables 8 and 9 from the previous version. 

Additional references to consider: - On time preferences: https://mpra.ub.uni-muenchen.de/103660/

Response from authors: This is an interesting paper but since we are not using hypothetical questions, we are not too sure where to fit the reference in. Also, the reference list is on the long side already, so we are reluctant to make it even longer.

6. PLOS authors have the option to publish the peer review history of their article (what does this mean?). If published, this will include your full peer review and any attached files. Do you want your identity to be public for this peer review? For information about this choice, including consent withdrawal, please see our Privacy Policy.

Reviewer #1: No

Reviewer #2: No

---

## [Decision Letter · Decision Letter 1]

21 Sep 2021

Mind Training, Stress and Behaviour - a Randomised Experiment

PONE-D-21-20053R1

Dear Dr. Belot,

We’re pleased to inform you that your manuscript has been judged scientifically suitable for publication and will be formally accepted for publication once it meets all outstanding technical requirements.

Kind regards,

Pablo Brañas-Garza, PhD Economics

Academic Editor

PLOS ONE

Additional Editor Comments (optional):

Reviewers' comments:

Reviewer's Responses to Questions

**Comments to the Author**

1. If the authors have adequately addressed your comments raised in a previous round of review and you feel that this manuscript is now acceptable for publication, you may indicate that here to bypass the “Comments to the Author” section, enter your conflict of interest statement in the “Confidential to Editor” section, and submit your "Accept" recommendation.

Reviewer #1: All comments have been addressed

Reviewer #2: All comments have been addressed

2. Is the manuscript technically sound, and do the data support the conclusions?

Reviewer #1: Yes

Reviewer #2: Yes

3. Has the statistical analysis been performed appropriately and rigorously? 

Reviewer #1: Yes

Reviewer #2: Yes

4. Have the authors made all data underlying the findings in their manuscript fully available?

Reviewer #1: Yes

Reviewer #2: Yes

5. Is the manuscript presented in an intelligible fashion and written in standard English?

Reviewer #1: Yes

Reviewer #2: Yes

6. Review Comments to the Author

Reviewer #1: I am satisfied with the authors' responses to my comments. I congratulate the authors on their nice work.

Reviewer #2: (No Response)

7. PLOS authors have the option to publish the peer review history of their article (what does this mean?). If published, this will include your full peer review and any attached files.

Reviewer #1: No

Reviewer #2: No

---

## [Editor Report · Acceptance letter]

21 Oct 2021

PONE-D-21-20053R1 

Mind Training, Stress and Behaviour - a Randomised Experiment 

Dear Dr. Belot:

I'm pleased to inform you that your manuscript has been deemed suitable for publication in PLOS ONE. Congratulations! Your manuscript is now with our production department. 

Kind regards, 

on behalf of

Dr Pablo Brañas-Garza 

Academic Editor

PLOS ONE